# Development of *Nervilia fordii* Extract-Loaded Electrospun PVA/PVP Nanocomposite for Antioxidant Packaging

**DOI:** 10.3390/foods10081728

**Published:** 2021-07-27

**Authors:** Peng Wen, Teng-Gen Hu, Yan Wen, Ke-Er Li, Wei-Peng Qiu, Zhi-Lin He, Hong Wang, Hong Wu

**Affiliations:** 1College of Food Science, Guangdong Provincial Key Laboratory of Food Quality and Safety, South China Agricultural University, Guangzhou 510642, China; pwen@scau.edu.cn (P.W.); 201822010208@stu.scau.edu.cn (K.-E.L.); 201822010215@stu.scau.edu.cn (W.-P.Q.); 201822010206@stu.scau.edu.cn (Z.-L.H.); 2School of Food Science and Engineering, South China University of Technology/Guangdong Province Key Laboratory for Green Processing of Natural Products and Product Safety, Guangzhou 510640, China; hu.tenggen@gdaas.cn (T.-G.H.); ShakingW@aliyun.com (Y.W.); 3Sericultural & Agri-Food Research Institute, Guangdong Academy of Agricultural Sciences/Key Laboratory of Functional Foods, Ministry of Agriculture/Guangdong Key Laboratory of Agricultural Products Processing, Guangzhou 510640, China

**Keywords:** electrspinning, *Nervilia fordii* extract, nano-encapsulation, antioxidant activity

## Abstract

An ethyl acetate extract from of *Nervilia fordii* (NFE) with considerable suppression activity on lipid peroxidation (LPO) was first obtained with total phenolic and flavonoid contents and anti-LPO activity (IC_50_) of 86.67 ± 2.5 mg GAE/g sample, 334.56 ± 4.7 mg RE/g extract and 0.307 mg/mL, respectively. In order to improve its stability and expand its application in antioxidant packaging, the nano-encapsulation of NFE within poly(vinyl alcohol) (PVA) and polyvinyl(pyrrolidone) (PVP) bio-composite film was then successfully developed using electrospinning. SEM analysis revealed that the NFE-loaded fibers exhibited similar morphology to the neat PVA/PVP fibers with a bead-free and smooth morphology. The encapsulation efficiency of NFE was higher than 90% and the encapsulated NFE still retained its antioxidant capacity. Fourier transform infrared spectroscopy (FTIR) and X-ray powder diffraction (XRD) analysis confirmed the successful encapsulation of NFE into fibers and their compatibility, and the thermal stability of which was also improved due to the intermolecular interaction demonstrated by thermo gravimetric analysis (TGA). The ability to preserve the fish oil’s oxidation and extend its shelf-life was also demonstrated, suggesting the obtained PVA/PVP/NFE fiber mat has the potential as a promising antioxidant food packaging material.

## 1. Introduction

Antioxidant packing is a promising approach to avoiding oxidative damage of fatty foods by the incorporation of antioxidant agents into the packaging material [1]. This allows sustained release behavior of encapsulated antioxidants with high bioactivity, thus preserving food quality and extending its shelf-life. The increasing demand for healthy and safe food has promoted the study and the utilization of natural extracts, such as *allium ursinum L.* extract [2], *coptis chinensis* extract [3] and cinnamon essential oil [4]. *Nervilia fordii* (Hance) Schltr (Orchidaceae), also called “Qing Tian Kui”, is one kind of *Nervilia* species that is mainly distributed in the Southwest of China and Southeast Asia [5]. It has been employed as a traditional Chinese herbal medicine for the treatment of tuberculosis, cough, and throat swelling. Phytochemical studies have focused on the isolation and identification of bioactive compounds [6]. This is the first report regarding the antioxidant activity of different solvent extracts from *Nervilia fordii* and its potential for antioxidant packaging.

Studies have shown that the nanostructure of packaging material with a high surface area is beneficial for the release of encapsulated compounds compared to that of a bulk material. Electrospinning is an emerging and promising technique to produce nanofibers due to the ease of operation, versatility at room temperature, controllable fiber morphology, and high entrapment capacity of different bioactive compounds [7]. It has been well established in food areas such as active food packaging, targeted delivery, and functional foods [8,9,10]. Recently, more attention has been paid to biodegradable polymers because of the serious environmental problem caused by traditional plastic waste. Polyvinylpyrrolidone (PVP), a nontoxic and biocompatible polymer with excellent film-forming ability, has been applied as a packaging material [11]. Previous study showed that electrospun PVP nanofibers can be used to increase the solubility and achieve the controlled release of curcumin [12]. However, the neat PVP film is too brittle and easily broken, which limits its application. Poly(vinyl alcohol) (PVA) is also an electrospinable and non-toxic polymer permitted for food contact materials [13]. It has been reported that PVA has been blended with other polymers to encapsulate bioactive compounds, including chitosan [14], polycaprolactone [15], and soy protein isolate [16]. To the best of our knowledge, studies on combinations of PVA and PVP for the preparation of electrospun antioxidant packaging material have not been reported yet.

Hence, in this study, different solvent extracts of *Nervilia fordii* were prepared and the antioxidant activity was measured using DPPH radical scavenging and lipid peroxidation assay. After that, the development of the PVA/PVP-based biodegradable polymer matrix for the encapsulation of ethyl acetate extract of *Nervilia fordii* (NFE) using electrospinning was performed to improve its stability and antioxidant bioactivity. The morphology, encapsulation efficiency and structural characterization of the obtained composite fibers were investigated. Finally, the applicability for lessening the fish oil oxidation was also explored. 

## 2. Materials and Methods

### 2.1. Materials

The dried aerial part of *Nervilia fordii* with water content being around 11% was purchased from the Dashenlin Pharmaceutical Co. Ltd. (Guangzhou, China) with phenological stage being September in 2018; 2, 2-Diphenyl-1-picrylhydrazine (DPPH), ascorbic acid (Vc), rutin, gallic acid were obtained from Sigma-Aldrich Co. (Shanghai, China). PVP of molecular weight 1,300,000 was obtained from Aladdin Biological Technology Co., Ltd. (Shanghai, China); PVA (M w 85,000−124,000) was provided by Tianyi company (Guangzhou, China). All other reagents used were analytical grade.

### 2.2. Preparation of Different Solvent Extracts

The different *Nervilia fordii* extracts were prepared as follows: 10 g of air-dried aerial part of *Nervilia fordii* plant was mixed with 100 mL of a certain concentration of ethanol solution. The mixture was stirred at a certain temperature (50~90 °C) for 2 h, and the residues were re-extracted twice. Filtrates were pooled, and, after the evaporation of ethanol, the concentrated ethanol extract was then extracted with petroleum ether, ethyl acetate, and n-butanol for 4 h, respectively. The supernatants were combined and the residue was re-extracted by repeating the above procedure twice. The effects of variables like the ethanol concentration, temperature and solid-to-solvent ratio on DPPH activity were investigated. Finally, the obtained extracts were freeze-dried and kept in a desiccator for further analysis. 

### 2.3. Determination of Antioxidant Contents in Nervilia fordii Extracts

(1)Total phenol content (TPC)

TPC was measured by Folin–Ciocalteu Reagent assay according to a previous study with some modifications [17]. In brief, 100 μL of different concentrations of solvent extracts were individually dissolved in 70% ethanol. Then, 0.5 mL of Folin–Ciocalteu Reagent and 7.9 mL of water were added, followed by shaking. After incubation in the dark for 5 min, the solution was reacted with 1.5 mL of 10% Na_2_CO_3_ solution and incubated in the dark for 2 h. After that, the absorbance at 765 nm against a blank was measured. A series of gallic acid standard solution was used to establish a calibration curve. The results were expressed as mg gallic acid equivalent (GAE)/g extract. For the NFE-loaded film, the sample was first immersed into 10 mL of 70% ethanol. The obtained extract solution from film was then determined by the above method. The TPC is expressed as mg of GAE/g film.

(2)Total flavonoid content (TFC)

TFC was calculated by AlCl_3_-HAc-NaAc (pH 5.5) assay. Briefly, 200 μL different concentrations of solvent extracts were individually dissolved in 70% ethanol. Then 300 μL of 0.1 mol/L AlCl_3_ solution and 200 μL of HAc-NaAc buffer solution (pH 5.5) were serially added and mixed with 4.5 mL of 70% ethanol. The absorbance at 405 nm was monitored. A series of rutin standard solution was used to establish a calibration curve. The results are expressed as mg rutin equivalent (RE)/g extract.

### 2.4. Antioxidant Activity Analysis

(1)DPPH assay

DPPH assay was used to evaluate the free radical scavenging activity of each sample as described by Neo with slight modifications [18]. Briefly, a certain amount of different extracts was dissolved in 70% ethanol aqueous solutions. One milliliter of the sample solution was then mixed with 3 mL of 0.1 mM DPPH that dissolved in ethanol in the dark for 30 min. The DPPH radical-scavenging rate was expressed as follows:DPPH radical-scavenging rate % = (A_517, control_ − A_517,sample_)/A_517,control_ × 100%

A_517,sample_ and A_517,control_ are the absorbance of DPPH solution with or without samples. 

(2)Lipid peroxidation (LPO) assay

First, the yolk lipoprotein solution was prepared as follows: albumen was removed from fresh eggs and the yolk was mixed with the same volume of 0.1 mol/L phosphate buffer (pH 7.4). After shaking for 10 min, the mixture was diluted into 1:25 using phosphate buffer. The obtained solutions were kept at 4 °C. Then, 0.2 mL of the above solutions was serially mixed with 0.1 mL of tested sample, 0.2 mL of FeSO_4_ and 1.7 mL phosphate buffer. The mixtures were incubated at 37 °C for 1 h followed by adding 0.5 mL of trichloroacetic acid (TBA, 20%, *w*/*v*) solution. After 10 min incubation, the mixture was centrifuged at 4000 rpm for 10 min. The supernatant (2 mL) was taken out and mixed with 1 mL of TBA (0.8%) in a boiling water bath for 15 min. After cooling down to 25 °C, the absorbance was monitored at 532 nm. The LPO suppression ratio was determined as follows: LPO suppression ratio (%) = ((A_control_ − A_sample_)/A_control_) × 100%

A_sample_ and A_control_ are the absorbance of solution with or without samples.

### 2.5. Encapsulation of Nervilia fordii Extract by Electrospinning

NFE was first prepared according to the optimized extract conditions described in Section 2.2. After that, the blending solution of PVA with PVP containing NFE was achieved as follows. First, 10%~15% (*w*/*v*) of PVA solution was prepared by stirring at 80 °C for 2 h. Then, the NFE-loaded PVP solution that was dissolved in 90% ethanol was added into the above PVA solution with a volume ratio of 1:1 to achieve a total polymer mass ratio and NFE of 12.5% and 4%, respectively. Finally, the electrospinning solution was injected into a syringe (21# needle), and the electrospinning process was conducted as a feed rate of 0.3 mL/h, voltage of 16 kV and distance of 14 cm. The encapsulation efficiency (EE) and loading capacity (LC) of NFE were determined based on the measurement of the amount of non-encapsulated extract. In brief, hexane was used to remove the free NFE from the electrospun nanofiber for 2 min, and the absorbance of obtained solution was measured at 472 nm. A series of NFE solution dissolved in hexane was used to establish a calibration curve. The EE and LC were determined as below: EE% = (theoretical mass of NFE − free mass of NFE)/theoretical mass of NFE ∗100
LC% = (theoretical mass of NFE − free mass of NFE)/the mass of fiber∗100

### 2.6. Characterization of the Electrospun Fiber Mat

*Scanning Electron Microscopy (SEM)*. The morphology of the obtained fiber was observed by a 3700 N scanning electron microscopy (SEM, Hitachi, Japan). The average diameter of fibers and the fibers size distribution was analyzed by Image-J software. 

*Fourier transform infrared Spectroscopy (FTIR)*. The interactions among PVA, PVP and NFE were investigated by FTIR spectroscopy (Bruker-VERTEX 70, Germany). The analysis was conducted under wave number of 3800–500 cm^−1^ and resolution of 4 cm^−1^. 

*X-ray diffraction pattern (XRD)*. The XRD pattern of the different composite fibers was recorded to examine the crystallography of the prepared films using a MiniFlex 600 diffractometer (RigaKu, Tokyo, Japan) with Cu-Kα radiation. Data were collected in the *2θ* range from 5° to 60° with a step of 0.02°.

*Thermogravimetric Analysis (**TGA).* The thermal property of the fiber mats was characterized using a TGA Q500 (TA Instruments, New Castle, DE, USA). The sample was heated from 25 °C to 700 °C with a heating rate of 20 °C/min under nitrogen gas atmosphere.

### 2.7. Oxidative Stability in Accelerated Storage Test

The oxidative stability of encapsulated fish oil was analyzed under storage conditions of 45 °C since low and ambient temperatures require a relative long period of time. As described previously [19], 10 mL of fish oil sample was aliquoted into 20 mL brown glass vials, and 20 mg TA fibrous mat (2 cm × 3 cm) were added, followed by incubation at 45 °C in the dark for 30 days, as shown in Figure 1. This subject is of primary importance, since the goal of the paper is to evaluate the potential application of the antioxidant film as a pouch or direct contacting material. The un-encapsulated fish oil was used as the control, while oil samples with pure PVA/PVP film were used as a negative control. Peroxide value (PV) analysis was conducted by taking different amounts of samples out at different intervals as follows [20]. Briefly, 3 g of fish oil sample was dissolved in 50 mL of acetic acid and chloroform mixture (3:2 *v*/*v*). Then, 1 mL of saturated KI solution was added and the mixture was kept in the dark for 1 min. After adding distilled water (50 mL), the mixture was immediately titrated with 0.01 mol/L of sodium thiosulfate until the yellow color had almost disappeared. The PV value was determined as follows: PV value (mEq of O_2_/kg sample) = 12.69 × 78.8 × (V_S_ − V_B_) × C/m.
where V_S_ and V_B_ is the volume of titrant used in the titration of oil sample and a blank without any oil sample (mL), respectively, C is the concentration of sodium thiosulfate (mol/L) and m is the weight of oil sample (g). 

### 2.8. Statistical Analysis

All experiments were performed in triplicate. The obtained results were reported as the mean values ± standard deviations. Significant differences were carried out by SPSS 17 statistical software (SPSS Inc., Chicago, IL, USA), as *p* < 0.05 is regarded to be significantly different.

## 3. Results

### 3.1. Preparation of Nervilia fordii Ethanol Extract

The extraction parameters for *Nervilia fordii* ethanol extract was first optimized using single factor experiments by investigating the effects of variables such as the ethanol concentration, temperature and solid-to-solvent ratio on the antioxidant activity and yields of *Nervilia fordii* ethanol extract. Regarding the ethanol concentration, from Figure 2a, it can be seen that no significant difference (*p* < 0.05) in DPPH scavenging rate between the two ethanol concentrations (60% and 70%) was observed. For the extraction temperature, it was found that the increase of temperature would cause an enhanced DPPH scavenging activity (Figure 2(b1)), which may be ascribed to the accelerated molecular movement and decreased solvent viscidity. There was no statistical significance (*p* > 0.05) among the extraction temperatures with the value of 70 °C vs. 80 °C. In addition, higher temperature could cause the degradation of the flavonoids and phenolic compounds, as exhibited in the yields (Figure 2(b2)), a similar phenomenon was reported by Altemimi et al. [21]. Figure 2(c1) shows that DPPH activity was significantly increased (*p* < 0.05) with the increase of the solid-to-solvent ratio from 1:5 to 1:15, and no significant difference (*p* < 0.05) was observed between 1:10 and 1:15. This finding agreed with Prasad et al. who found that the permeation of compounds into the solvent can be enhanced under the higher solid-liquid ratio [22]. However, too high of a liquid-solid ratio can restrain the cavitation effect (Figure 2(c2)). Hence, from an economic point of view, the optimized process for *Nervilia fordii* ethanol extract was: ethanol concentration 60%, extraction temperature 70 °C and solid-to-solvent ratio 1:10.

After that, Box-behnken design (BBD) and response surface methodology (RSM) were then adopted for the optimization of extraction parameter. The levels for response surface design are shown in Appendix A. The ANOVA results (Appendix A) revealed that the model was remarkably significant (*p* < 0.0001), and the R-squared value obtained was 0.9968, indicating the model was very consistent with the experiment results. The fitted equation was: DPPH scavenging rate (%) = 38.95 − 0.75*A + 1.53*B + 1.69*C + 0.31*B*C + 0.28*A*C − 0.25*A*B − 0.43*A^2^ − 0.81*B^2^ − 1.04*C^2^ (A—Ethanol concentration; B—Temperature; C—Solid to liquid ratio). Besides that, two-dimensional contour plots and three-dimensional response surface for the correlation between any two variables are displayed in Appendix A. The optimal condition for extraction of *Nervilia fordii* ethanol extract was obtained as follows: ethanol concentration 55.55%, extraction temperature 75 °C and solid–liquid rate 14.2:1. Regarding the practical situation, the optimal condition was adjusted to an ethanol concentration of 56%, extraction temperature of 75 °C and solid–liquid rate of 14:1. Under which, the DPPH scavenging rate of *Nervilia fordii* ethanol extract obtained was 41.5%, which is closer to the theoretical prediction value of 40.9%, suggesting that the model was desirable.

### 3.2. Antioxidant Capacities of Different Solvent Extracts

Then, different solvents (petroleum ether, ethyl acetate and n-butyl alcohol) were used to further extract the above obtained *Nervilia fordii* ethanol extract and their antioxidant activities were evaluated. From Figure 3a, it can be seen that the activity increased with the increase of extract concentration, and the scavenging abilities on DPPH radicals were in the order of ethyl acetate extract of *Nervilia fordii* (NFE) > n-butyl alcohol extract of *Nervilia fordii* (NFB) > petroleum ether extract of *Nervilia fordii* (NFP), which reached 94.5%, 60.5%, 33% and 18.4% at 3 mg/mL, respectively. However, a better scavenging rate was exhibited by Vc (96.3% at 0.1 mg/mL). As summarized in Table 1, the EC_50_ values of DPPH radical-quenching activity for NFE, NFB and NFP were 0.66, 2.43 and 4.25 mg/mL, respectively. Lipid peroxidation (LPO) is also considered to be another type of free radical oxidation, which is related to cellular damage. Similar to the tendency of DPPH scavenging activity of extracts, the LPO inhibition potency of different extracts were also concentration dependent and NFE possessed the highest inhibitory effect. Figure 3b shows that the inhibition rate rose from 24.9% to 77.4% for Vc and from 12.8% to 88.7% for NFE with the concentration increasing from 100 μg/mL to 500 μg/mL, respectively. In particular, a higher inhibition rate was achieved for NFE than that of Vc when the antioxidant concentration >300 μg/mL. From the IC_50_ values of LPO suppression activity in Table 1, NFE (0.307 mg/mL) has considerable activity in comparison to Vc (0.310 mg/mL), while NFB was 0.347 mg/mL and NFP was 0.436 mg/mL. These results clearly indicated that all of the solvent extracts had a noticeable effect on the inhibition of LPO, especially for the NFE, suggesting its potential in inhibiting the oxidation of fatty food or the application for functional foods.

### 3.3. Total Phenol and Flavonoid Contents

Being the main chain-breaking antioxidants that contributed to the antioxidant activity of extracts, the total phenolic and flavonoid contents in different extracts were examined using gallic acid equivalents (GAE) and rutin equivalents (RE), respectively. As presented in Table 1, the NFE had significantly higher contents of total phenols (86.67 ± 2.5 mg GAE/g extract) and total flavonoids (334.56 ± 4.7 mg RE/g extract), followed by NFB and NFP. It can be concluded that the polar solvent extracts (n-butanol and ethyl acetate) exhibited a higher content than nonpolar solvent extracts (petroleum ether) did, which may contribute to the stronger antioxidant activity of NFE and NFB than that of NFP. Taken the polarities of used solvents into consideration, it has been revealed that flavonoids from *Nervilia fordii* were more extractable by solvents with high polarity.

### 3.4. Encapsulation of NFE into PVA/PVP Electrospun Nanofibers

As shown in Figure 4, the different NFE content-loaded PVA/PVP fibers did not show different morphology compared to the neat PVA/PVP fibers. However, the diameters of nanofibers significantly increased with the addition of NFE, which could be due to the solution characteristics as shown in Table 2. Although the conductivity decreased with the increase of NFE into the PVA/PVP solution, no significant decrease was observed. The viscosity of the solution increased from 1535 Pa·S to 1693 Pa·S because of the molecule entanglements, which was favorable for the formation of thick fibers [23]. A similar phenomenon was observed in tomato peel extract-loaded gelatin fibers [24]. In addition to fiber diameter, the DPPH radical scavenging rate also enhanced by increasing NFE concentration, suggesting the potential for antioxidant packaging (Table 2). When the NFE concentration was 4.0 mg/mL, the scavenging rate reached 78.4%, and there was no significant difference (*p* > 0.05) compared to that of 8 mg/mL. Besides that, the encapsulation efficiency values for 2 mg/mL, 4 mg/mL and 8 mg/mL NFE were 96.57 ± 1.46%, 94.32 ± 1.78%, and 91.42 ± 2.45%, respectively. These results suggested that almost no loss of NFE occurred during the electrospinning and more than 90% of NFE could be encapsulated into PVA/PVP fibers, indicating the efficient nano-encapsulation of NFE by electrospinning.

### 3.5. Characterization of Electrospun Nanofibers

The interactions and compatibility among different components were examined by FTIR spectroscopy. Figure 5 shows that NFE exhibited peaks of O–H stretching (3330 cm^−1^), C=O stretching (1705 cm^−1^), and C–O–C) stretching (1242 and 1079 cm^−1^). Compared to neat PVP and PVP fibers, almost no changes in the spectra of PVP/NFE and PVA/NFE were observed, suggesting that NFE could be compatibly entrapped into the PVP and PVA fibers. When NFE was added to the PVA/PVP matrices, the infrared spectra of NFE-loaded PVA/PVP fiber mat exhibited the characteristic peaks of both polymers, while the presence of NFE peaks almost disappeared, indicating the encapsulation of NFE into the fiber mat due to the formation of various kinds of intra/inter-molecular hydrogen bonds. In particular, the intensity and width of the peak broadening over 3380 cm^−1^ was in the following order of PVA/PVP/NFE > PVA/PVP, which indicated that a stronger hydrogen bond was formed in PVA/PVP/NFE.

A crystallinity analysis by XRD pattern was carried out to confirm the inter-molecular interaction and biocompatibility among different components. As shown in Figure 6, pure NFE exhibited patterns of a crystalline state with diffraction peaks at 19.67°, 22.75° and 24.63°; however, these characteristic peaks for NFE were absent in the diffractograms of PVA/NFE, PVP/NFE and PVA/PVP/NFE. It can be suggested that NFE was converted into an amorphous state, which was ascribed to the electrospinning process and the interaction with amorphous PVP polymer. The rapid evaporation of solvent caused the quick transformation of polymer solution into solid form without enough time for crystallization [25]. Besides that, it can be seen that the first peak of PVP/NFE at 10.82° significantly diminished in PVP/PVA/NFE, and the intensity of the PVA/NFE peak at 19.47° was weakened and shifted to the higher 2θ position in PVP/PVA/NFE. These findings corresponded with the FTIR study where the intermolecular interaction existed among different components, particularly as hydrogen bonds.

TGA was used to evaluate the thermal stability of the obtained fiber mat. From Figure 7, the weight loss around 60~180 °C was attributed to the evaporation of free, freezing bound and chemical bound water for all curves. Free water is usually related to the absorption water and the freezing bound water is weakly interacted with the polymeric chain, while the chemical bound water is regarded as the water molecules bound to the polymeric chains via hydrogen bound. Thus, the mass loss in the first region of the thermograms related to different samples can be ascribed as evaporation of different types of water. For the PVA/NFE curve, another two main weight loss regions were displayed. The first region around 280~330 °C was due to the degradation of the side chain in the PVA matrix, like the C–O bond, and the second stage (400~460 °C) appeared as the cleavage of the C–C backbone in polymers, leading to so-called carbonization. In the case of PVP/NFE film, the region around 400~460 °C was attributed to the degradation of PVP. In particular, the higher degradation temperature is a result of increased number of hydroxyl groups among components. These findings are consistent with the previous results of FTIR and XRD analysis. Taking all results into consideration, it can be concluded that the obtained PVA/PVP/NFE fiber mat possesses adequate thermal stability and can be safely used for packaging application.

### 3.6. Effectiveness of the Active Films against Lipid Oxidation of Fish Oil

Oxidation is one of the major causes of oil deterioration. For example, fish oil, a nutritive functional food, is highly susceptible to oxidation due to the presence of polyunsaturated fatty acids, thus limiting its application. Hydroperoxides are the primary products formed in the first stage of oxidation, which can be reflected by peroxide values (PV), an indicator that represents the extent of early phase lipid oxidation [26]. For an edible food product, the PV levels should be less than 30 meq peroxide/kg oil [27]. Herein, the effect of different antioxidant films on the oxidation extent of fish oil was determined by measuring the PV at 25 °C for 60 days. Unpacked fish oil was used as blank. As depicted in Figure 8, the initial raw fish oil presented a PV value of 3.03 meq peroxide/kg oil. It was increased markedly for non-packed fish oil after day 10, reaching a maximum level of 378 meq peroxide/kg oil at day 50, and then declined to 343 meq peroxide/kg oil at day 60 (*p* > 0.05), which may be due to the degradation of peroxides as secondary oxidation began [28]. A similar trend was also noted for fish oil packed with PVP/PVA film, although there was a slight suppression in the PV values compared with the control, which could be related to the excellent gas barrier property of PVA-based films [29]. On the contrary, the different forms of antioxidants resulted in different extents of inhibitory effects on oxidation, as the PV values of samples packed with PVP/PVA/NFE were always lower than that of the oil treated with free NFE on all sampling days. For instance, the PV value (30 meq peroxide/kg oil) occurred on the 10th, 30th and 40th day with respect to the unpacked oil and oil packed with PVA/PVP film and free NFE, respectively. The results indicated that the incorporation of natural antioxidants into an electrospun nanoscale delivery system was able to maintain its activity and was more effective than that of the non-encapsulated antioxidant due to the improved solubility. A similar trend was found in the edible guar gum-based nanofibrous mat for the encapsulation of tannic acid to inhibit the oxidation of flaxseed oil [19].

## 4. Conclusions

In this study, the ethyl acetate extract of *Nervilia fordii* (NFE) possessing excellent antioxidant activity was first obtained, followed by encapsulation into a PVA/PVP nanofiber with high encapsulation efficiency. FTIR-ATR, TGA and XRD analysis confirmed the presence of hydrogen bonding interactions among different components and their miscibility at the molecular level. The prepared nanofiber still exhibits good antioxidant activity by inhibiting the primary oxidation products of fish oil, in particular, it was more effective than the non-encapsulated NFE. These results suggested that nanoencapsulation by electrospinning is an effective way to stabilize NFE and improve its antioxidant activity. Hence, the PVA/PVP/NFE nanofiber could be a promising antioxidant packaging material for fatty food preservation.

## Figures and Tables

**Figure 1 foods-10-01728-f001:**
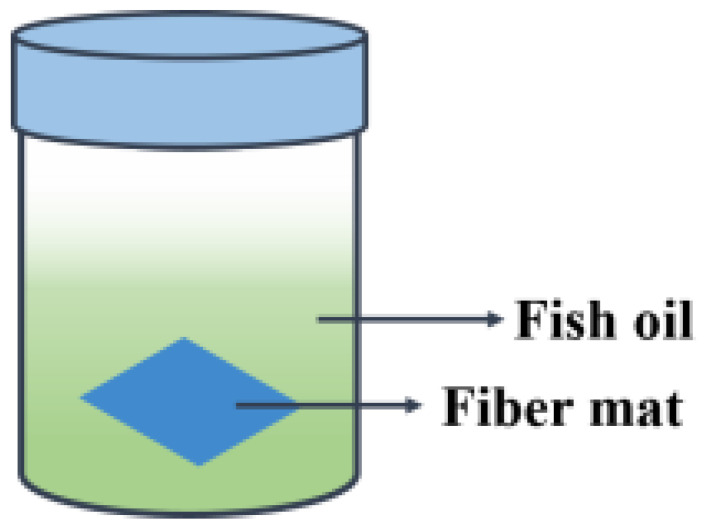
The schematic diagram for measuring the fiber mat’s antioxidant activity on fish oil oxidation.

**Figure 2 foods-10-01728-f002:**
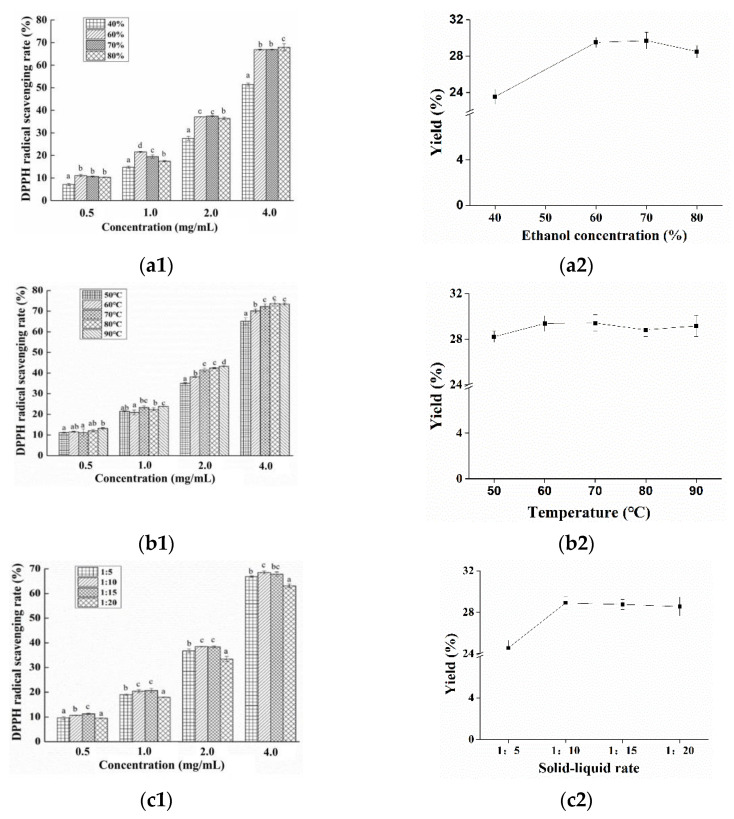
Effects of extraction condition on the extract’s DPPH radical scavenging activity and yield. ((**a1**–**c1**) indicated the effects of ethanol concentration, temperature and solid-liquid rate on the DPPH radical scavenging rates of *Nervilia fordii* ethanol extract, respectively; (**a2**–**c2**) indicated the effects of ethanol concentration, temperature and solid-liquid rate on the yields of *Nervilia fordii* ethanol extract, respectively).

**Figure 3 foods-10-01728-f003:**
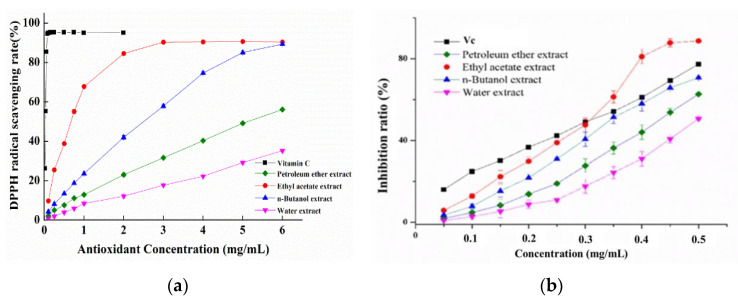
The DPPH radical scavenging ability (**a**) and the LPO suppression ratio (**b**) of different extracts and Vc.

**Figure 4 foods-10-01728-f004:**
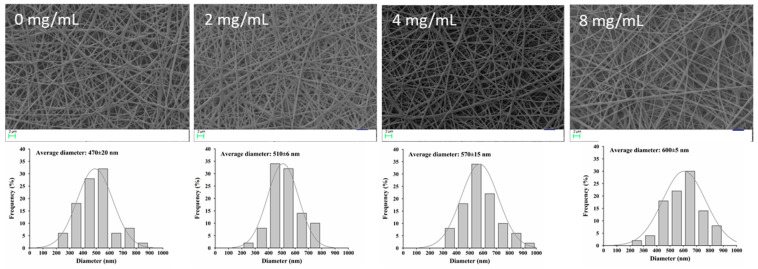
SEM images of different NFE content-loaded PVA/PVP fibers and its diameter distribution.

**Figure 5 foods-10-01728-f005:**
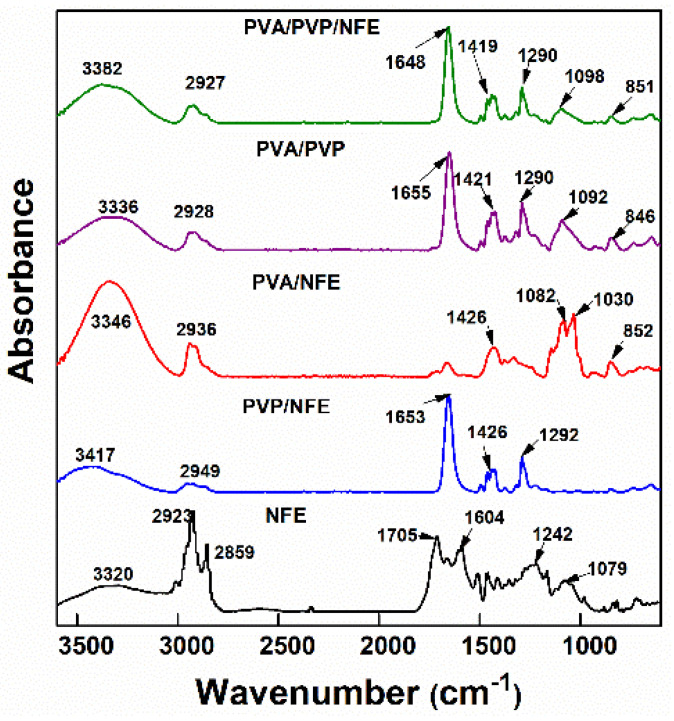
FTIR of different samples.

**Figure 6 foods-10-01728-f006:**
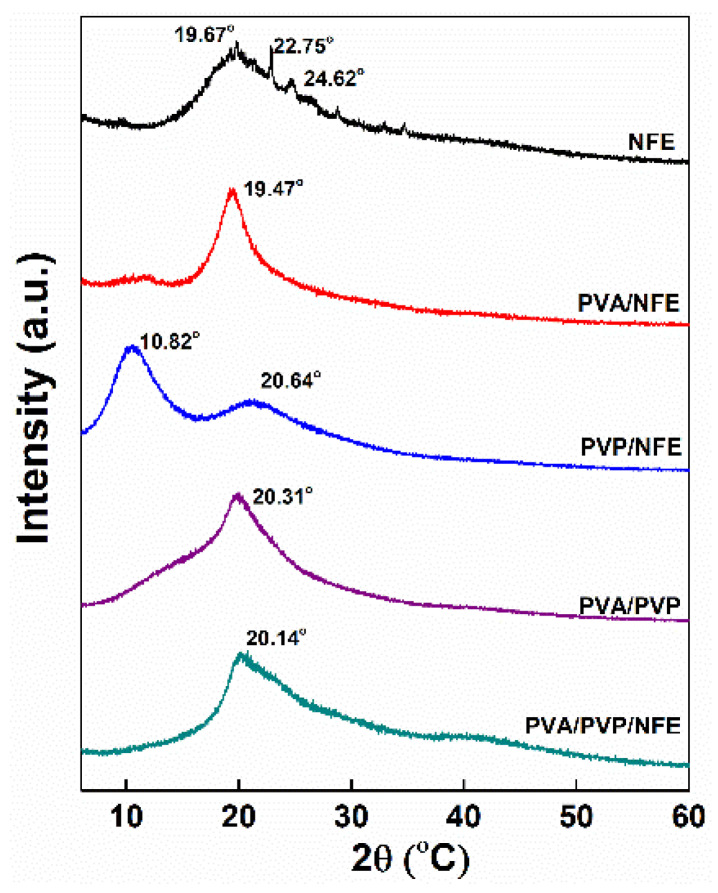
X-ray diffractograms of different samples.

**Figure 7 foods-10-01728-f007:**
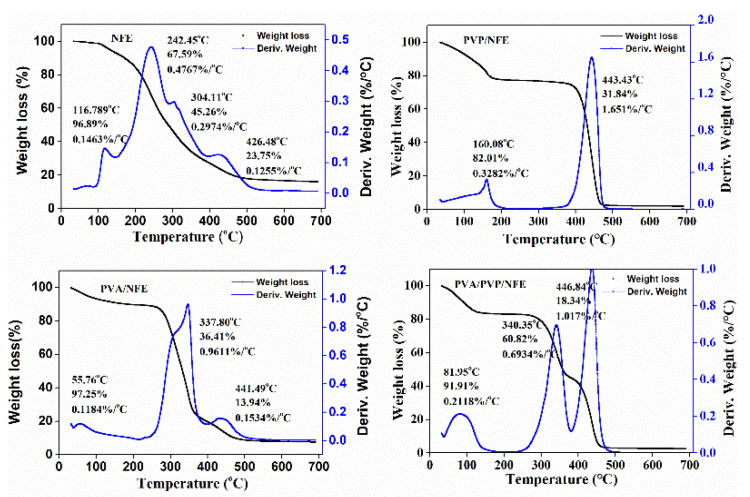
TGA curves of different samples.

**Figure 8 foods-10-01728-f008:**
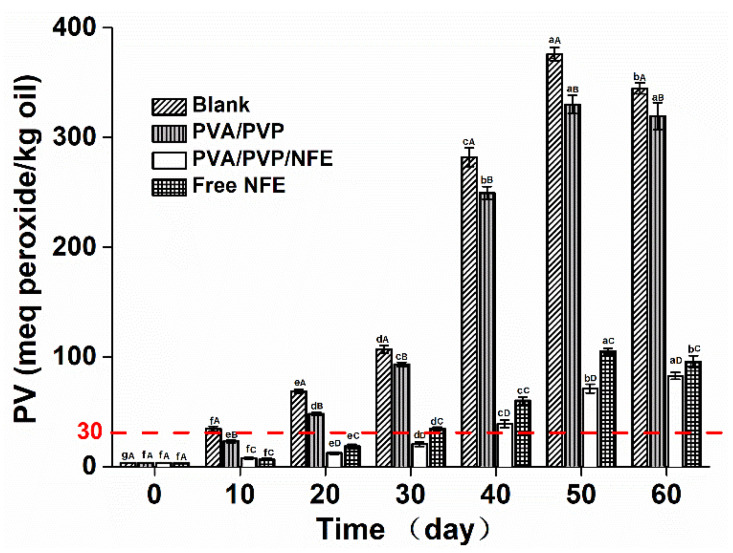
X-ray diffractograms of different samples. Different lowercase letters indicated a significant difference (*p* < 0.05) over different incubation time that treated with the same film. Different capital letters indicated a significant difference (*p* < 0.05) among different packaging film at a specific incubation time.

**Table 1 foods-10-01728-t001:** Extraction yields, and total phenols, total flavonoid contents in different extracts.

Sample	Content		EC50 ^c^ of DPPH	IC50 ^d^ of LPO
	Total Phenols (mg GAE ^a^/g Extract)	Total Flavonoids (mg RE ^b^/g Extract)	Radical-Quenching Activity (mg Sample/mL)	Suppression Activity (mg Sample/mL)
Ethyl acetate extract (NFE)	86.67 ± 2.5	334.56 ± 4.7	0.66	0.307
n-Butyl alcohol extract (NFB)	41.85 ± 1.3	125.9 ± 2.6	2.43	0.347
Water extract	18.04 ± 1.2	46.95 ± 1.5	8.36	0.492
Petroleum ether extract (NFP)	15.17 ± 0.8	100.36 ± 3.1	4.25	0.436
Vc			0.087	0.310

Values (mean ± SD, *n* = 3) in the same column followed by a different letter are significantly different (*p* < 0.05). ^a^ GAE, Gallic acid equivalents. ^b^ RE, Rutin equivalents. ^c^ EC50 means the effective concentration of sample that can decrease 50% of DPPH radical scavenging rate. ^d^ IC50 means the effective concentration of sample that can inhibit 50% of lipid peroxidation.

**Table 2 foods-10-01728-t002:** Characteristics of electrospun solutions and the obtained fibers.

NFE Concentration (mg/mL)	Viscosity (Pa·S)	Conductivity (μS/cm)	Average Diameter (nm)	DPPH Radical Scavenging Rate (%)
0	1535	326	470	0 ^a^
2.0	1572	315	510	38.7 ± 1.2 ^b^
4.0	1604	307	570	78.4 ± 2.4 ^c^
8.0	1693	302	600	80.2 ± 1.9 ^c^

Note: different letters in the same column indicates the statistically different (*p* < 0.05).

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
