# Peer review of "Development of Nervilia fordii Extract-Loaded Electrospun PVA/PVP Nanocomposite for Antioxidant Packaging"

_foods, 2021, doi:10.3390/foods10081728_

Round 1
Reviewer 1 Report
The manuscript needs of some minor review.
This is an interesting research topic, and it is clear that a good work has been done.
The manuscript is well documented with valid bibliographical references. However, results discussion should be improved, and some specific changes should be made:
Line 21-23 - Abbreviations should only be used after being spelled out for the first time
Line 200 - correct reference name, should be Prasad et al.
Line 353 - Correct figure number
Author Response
Dear Editor/Reviewers:
We are very grateful for your helpful suggestions for revision (Manuscript ID: FOODS_1295078). The manuscript has been carefully revised according to the editor and reviewers’ comments. All the changes are marked in red fonts throughout the revised manuscript. The issues raised by the editor and reviewers have been addressed as follows. Page, line or figure numbers refer to the revised manuscript.
Point 1: Line 21-23 - Abbreviations should only be used after being spelled out for the first time Response 1:As suggested by the reviewer, the abbreviations had been spelled out for the first time in the revised manuscript. (P1 li.21-25)
Point 2: Line 200 - correct reference name, should be Prasad et al. Response 2:The author name had been revised in the revised manuscript. (P5 li.203)
Point 3: Line 353 - Correct figure number Response 3:The figure number had been corrected in the revised manuscript. (P10 li.357)
Reviewer 2 Report
This study deals of the antioxidant efficiency against lipid peroxidation of antioxidant packaging developed with vegetable extracts (Nervilia fordii) and biodegradable polymers (PVA/PVP) on fish oil.
Some revisions need to be made, before a publication.
1)Material and methods:
a)Which part of Nervilia fordii plant , dry or fresh, was used for the preparation of the extracts?. The sampling is described with poor method (Section 2.2).
Results
2)Line 353, page 10: Ckeck the number of the Figure 6.
3) Applicability was studied only respect to a single primary oxidative parameter on a lipid food. Consider this one in the conclusions.
What type of bioactive compounds could be released by the biopolymer?
Moreover is it possible to apply this antioxidant packaging material even on hydrophilic food samples?
Round 2
Reviewer 2 Report
After the first revision, other improvements are needed:
1) Respect to the Authors’s revisions ‘The air-dried aerial part of Nervilia fordii plant was used for the preparation of the extracts. (P2 li.78-79), now right units of measurements have to be used for analyses in particular because of as reported in the manuscript after the revisions, there is a not congruence for the units obtained for different extracts. Revise please.
2) insert the average umidity and the right phenological stage of aerial part of Nervilia fordii plant used as samples.
3)After the revision in Conclusions: ‘Hence, the PVA/PVP/NFE nanofiber could be a promising 386 antioxidant packaging material in food preservation’ it is necessary to explain this observation in the conclusions, remember the use of PVA/PVP/NFE on in high fatty foods .
